# Identification of a reliable fixative solution to preserve the complex architecture of bacterial biofilms for scanning electron microscopy evaluation

**Rohana P. Dassanayake**[1]*, **Shollie M. Falkenberg**[1], **Judith A. Stasko**[2], **Adrienne L. Shircliff**[2], **John D. Lippolis**[1], **Robert E. Briggs**[1]

**1** Ruminant Diseases and Immunology Research Unit, National Animal Disease Center, United States Department of Agriculture, Agricultural Research Service, Ames, Iowa, United States of America, **2** Microscopy Services Laboratory, National Animal Disease Center, United States Department of Agriculture, Agricultural Research Service, Ames, Iowa, United States of America

* rohana.dassanayake@usda.gov

**Data Availability Statement:** All relevant data are within the paper.

**Funding:** This research was conducted at a USDA research facility and all funding was provided

## Abstract

Bacterial biofilms are organized sessile communities of bacteria enclosed in extracellular polymeric substances (EPS). To analyze organization of bacteria and EPS in high resolution and high magnification by scanning electron microscopy (SEM), it is important to preserve the complex architecture of biofilms. Therefore, fixation abilities of formalin, glutaraldehyde, and Methacarn (methanol/chloroform/acetic acid-6:3:1) fixatives were evaluated to identify which fixative would best preserve the complex structure of bacterial biofilms. Economically important Gram-negative *Mannheimia haemolytica*, the major pathogen associated with bovine respiratory disease complex, and Gram-positive *Staphylococcus aureus*, the major cause of chronic mastitis in cattle, bacteria were selected since both form biofilms on solid-liquid interface. For SEM analysis, round glass coverslips were placed into the wells of 24-well plates and diluted *M. haemolytica* or *S. aureus* cultures were added, and incubated at 37°C for 48–72 h under static growth conditions. Culture media were aspirated and biofilms were fixed with an individual fixative for 48 h. SEM examination revealed that all three fixatives were effective preserving the bacterial cell morphology, however only Methacarn fixative could consistently preserve the complex structure of biofilms. EPS layers were clearly visible on the top, in the middle, and in the bottom of the biofilms with Methacarn fixative. Biomass and three-dimensional structure of the biofilms were further confirmed spectrophotometrically following crystal violet staining and by confocal microscopy after viability staining. These findings demonstrate that Methacarn fixative solution is superior to the other fixatives evaluated to preserve the complex architecture of biofilms grown on glass coverslips for SEM evaluation.

through internal USDA research dollars. This project is an intramural project of the USDA/ARS National Animal Disease Center (5030-32000-116-00D). The funders had no role in study design, data collection and analysis, decision to publish, or preparation of the manuscript.

**Competing interests:** The authors have declared that no competing interests exist.

## Introduction

Bovine respiratory disease complex (BRDC) and mastitis are the two major diseases affecting cattle associated with extensive economic losses to the US beef and dairy cattle industries [1–4]. BRDC is a multifactorial disease and develops as a result of an interplay between multiple viral and bacterial infections, stress, immune dysfunction, and environmental factors [1, 5]. Commensal bacteria such as *Mannheimia haemolytica*, *Pasteurella multocida*, *Histophilus somni*, and *Mycoplasma bovis* are commonly found in the upper respiratory tract of healthy cattle [1, 6]. Among them, *M. haemolytica* has been identified as the leading bacterial pathogen associated with BRDC [7, 8]. Although a variety of bacterial pathogens are known to cause mastitis, *Staphylococcus aureus* is one of the leading bacteria causing acute, subacute, subclinical, and chronic mastitis in lactating dairy cows [9–12]. It has been reported that naturally occurring chronic subclinical *S. aureus* mastitis in cows have the lowest cure rate of the major bacterial mastitis agents after antibiotic treatment [13]. The low mastitis cure rate may be due to the ability of *S. aureus* to invade mammary alveolar cells, macrophages, and/or the formation of biofilms [14, 15].

Biofilms are organized sessile multicellular communities of microbes encased in self-produced highly structured extracellular polymeric substances (EPS) or matrix [16, 17]. Microcolonies in the biofilms are separated by open water channels. EPS consists of extracellular polysaccharides, nucleic acids (DNA and RNA), and proteins; however, up to 97% of EPS can be accounted for by water [18, 19]. Additionally, peptidoglycan, lipids, phospholipids, and other bacterial cell components can be found in the EPS [18]. Bacteria can form biofilms on biotic or living cell surfaces (such as extracellular matrix protein molecules of cells) and also on abiotic or inert surfaces (such as plastics, glasses, metals, soil etc.). Biofilms can be formed at the air-liquid as well as solid-liquid interfaces [20, 21]. Many bacteria exist on or inside the host as part of biofilms, and recurrent infections (~80% infections in human) are often associated with the growth of bacteria from the biofilms [22]. The ability of microbes to form biofilms is a significant survival mechanism and therefore, unlike planktonic bacteria in the supernatants, sessile bacteria in biofilms show increased resistance to antimicrobial agents [22, 23].

Biofilm formation by *Mannheimia haemolytica* strains *in vitro* on abiotic (polystyrene plastic) and biotic (cultured bovine respiratory epithelial cells) surfaces has been previously reported [24, 25]. Long-term colonization of *M. haemolytica* in deep tonsillar crypts is thought to be due to the formation of biofilms. To support this notion, one case study demonstrated the presence of *M. haemolytica* in biofilm-like microcolonies embedded within the amorphous bacterial glycocalyx from a BRDC affected bovine lung tissue sample [26]. Similarly, various *Staphylococcus aureus* isolates, including the ones causing mastitis in lactating dairy cows, can also form biofilms *in vitro* on abiotic (polystyrene plastic) as well as biotic (cultured mammary epithelial cells and HeLa cells) surfaces [9, 27, 28]. Recurrent and chronic mastitis caused by *S. aureus* has been attributed to the growth of bacteria in biofilms of affected cows [15, 29]. It is important to highlight that both of these bacteria form biofilms at the solid-liquid interface.

Although spatial organization of bacterial cells and EPS in the biofilms can be determined by confocal laser scanning microscopy, only SEM can show the architecture of biofilms at a single bacterial cell resolution level. However, if a suitable fixative is not selected prior to SEM, three-dimensional structures of biofilms may be damaged. Therefore, the goal of the current study was to identify a superior fixative solution that can best preserve the complex architecture of biofilms to be evaluated under SEM. Such preservation allows us not only to analyze sessile bacterial cells in microcolonies of the biofilms, but also to visualize EPS layer organization within the biofilms.

Fixatives can be broadly classified into two categories: cross-linking fixatives (10% formalin or neutral-buffered formalin (NBF), glutaraldehyde, and paraformaldehyde) and denaturing fixatives (B5, Carnoy's, Methacarn, and Zenker's) [30]. The commonly used fixatives to preserve bacterial biofilms include NBF and 2.5% glutaraldehyde. EPS is a major contributor to the formation of the complex architecture of biofilms. Furthermore, the majority of total organic carbon was found in the EPS as compared to bacterial cell biomass [31, 32]. However, biofilms fixed in NBF and glutaraldehyde on abiotic surfaces often show smaller areas of EPS layers as compared to well-preserved sessile bacterial microcolonies [25, 33–35]. Carnoy's fixative solution (ethanol/chloroform/acetic acid-6:3:1) has been commonly used to preserve mucus since fixation with formalin often causes the loss of mucus layers [36, 37]. These studies also demonstrated that formalin fixation lead to loss of biofilms. It is our understanding that Carnoy's or Methacarn (Methanol-Carnoys; methanol/chloroform/acetic acid-6:3:1) fixative solution has not been widely tested as a fixative to preserve the bacterial biofilms grown on abiotic surfaces. Therefore, fixation abilities of NBF, glutaraldehyde, and Methacarn fixative solutions were evaluated to determine which fixative would best preserve the complex structure of biofilms, in particular EPS layers, when examined under SEM.

## Materials and methods

### Bacterial strains, culture media, and growth conditions

*Mannheimia haemolytica* serotype 1 D153, a pathogenic strain associated with bovine respiratory disease complex [38] and *Staphylococcus aureus* Newbould 305 (NB305), a pathogenic strain associated with bovine mastitis [39] were maintained as frozen stocks (-80°C) in brain heart infusion broth (BHI; Cat#: 211059; Becton, Dickinson Co., Sparks, MD) with 10% glycerol (Cat#: G5516; Sigma-Aldrich, St Louis, MO). *M. haemolytica* was cultured on trypticase soy agar supplemented with 5% defibrinated sheep blood plates (Cat#: 221261; TSA II™, Becton, Dickinson Co.) at 37°C in a humidified atmosphere of 7.5% $CO_2$ for overnight (~18 h). *M. haemolytica* colonies were transferred to 10 ml BHI broth in a 50 ml conical tube, optical density was adjusted ($OD_{600}$ = ~0.130), and incubated at 37°C with constant shaking (190 rpm) for about 2 h ($OD_{600}$ = 0.6; ~1 × $10^9$ colony forming units/ml [cfu/ml]). A loopful of *S. aureus* from a frozen stock was transferred into 10 ml BHI broth in a 50 ml conical tube and incubated overnight (~18 h,) at 37°C with constant shaking (200 rpm; ~5 × $10^8$ cfu/ml).

### Quantification of biofilm biomass

Crystal violet staining method was used to quantify the total amount of biofilm biomass produced by *M. haemolytica* D153 and *S. aureus* NB305 on 24-well plates [24, 34]. Biofilms were allowed to form on 24-well clear, flat-bottom tissue culture-treated polystyrene plates (Cat# 3524; Corning Inc., Kennebunk, ME) as described previously but with some modifications (Boukahil and Czuprynski, 2015; Kong et al., 2018). *M. haemolytica* grown in BHI broth for 2 h ($OD_{600}$ = 0.6, mid log phase) was separately diluted in BHI broth and Gibco colorless RPMI 1640 medium (Cat#: 11835; ThermoFisher Scientific, Grand Island, NY, 1:1000, ~1 × $10^6$ cfu/ml). Overnight grown *S. aureus* was also separately diluted in BHI broth and RPMI 1640 medium (1:500, ~1 × $10^6$ cfu/ml). Approximately 800 μl of diluted *M. haemolytica* or *S. aureus* cultures were separately added into wells in the 24-well plates (6 wells per each bacteria and 6 wells per each medium). Similar volumes of BHI broth and RPMI 1640 medium without bacteria were added to wells and used as negative controls. Plates were covered with the lids. *M. haemolytica* was incubated at 37°C in a humidified atmosphere of 7.5% $CO_2$ for 72 h without shaking while *S. aureus* was incubated at 37°C for 48 h without shaking, to allow biofilms to develop. Culture media were aspirated and the biofilms allowed to air dry at room temperature

for 1 h. *M. haemolytica* biofilms in plates were heat fixed at 80˚C for 30 min then allowed to cool to room temperature. *S. aureus* biofilms were fixed with 99% (v/v) methanol (Cat#: 230–4; Honeywell Burdick and Jackson, Muskegon, MI) for 6 min then air dried. Following fixation, biofilms were incubated with filter sterilized 0.04% (w/v) crystal violet (Cat#: C-581; Fisher Chemical, Fair Lawn, NJ) in deionized water (800 μl/well) at room temperature for 30 min in the dark. Plates were washed two to three times with water then 800 μl of 33% (v/v) glacial acetic acid (Cat#: A38S; Fisher Chemical) in water was added into each well. Solubilized biofilms in each well were transferred to 96-well clear, flat-bottom polystyrene plates (Cat#: 3596; Corning Inc.; 200 μl/well, quadruplicate) and the optical densities (OD) were recorded at 630 nm ($OD_{630}$) using a microplate reader (FlexStation 3, Molecular Devices LLC., San Jose, CA). Each assay was performed three times with six replicate wells per each sample.

## Confocal laser scanning microscopy (CLSM)

*M. haemolytica* D153 and *S. aureus* NB305 biofilms were allowed to form in Immulon 2HB 96-well clear, flat-bottom polystyrene plates (Cat# 3455; Thermo Scientific, Rochester, NY) and Ibidi 96-well optical bottom (polymer coverslip) μ-plates (Cat# 89626; Ibidi USA, Inc., Fitchburg, WI) as described for 24-well plates under the quantification of biofilm biomass. However, only 100 μl (Immulon 2HB) to 200 μl (Ibidi μ-plates) of diluted *M. haemolytica* and *S. aureus* each diluted in BHI broth and RPMI 1640 medium were separately added into each well in the 96-well plates. Plates were covered with plate sealers (ThermoFisher Scientific) or lids and incubated at 37˚C for 48 h– 72 h as described before. Biofilms were stained with Filmtracer™ LIVE/DEAD™ Biofilm Viability Kit (Cat#: L10316; ThermoFisher Scientific) containing SYTO 9 and propidium iodide (PI). Briefly, working staining solution was prepared by adding 3 μl of SYTO 9 and 3 μl PI into 1.2 ml of Dulbecco's phosphate buffered saline, pH 7.2 (Cat#: 14190; calcium chloride and magnesium chloride free; ThermoFisher Scientific). One hundred μl of staining solution was carefully added into each well then the plates were incubated at room temperature for 15 min in the dark. Live and dead bacteria in the biofilms were visualized using a Nikon A1R+ Confocal System microscope (Nikon Instruments, Melville, NY). SYTO 9 and PI were excited by 488 nm and 561 nm solid state laser beams and emission signals between 500–550 nm (green, live bacteria) and 570–620 nm (red, dead bacteria) were recorded, respectively. Images were obtained with plan Apo VC 20×/0.75 NA objective lens and plan Apo λ 60×/1.4 NA objective lens (oiled). All the *xy* and *z*-stack images were collected using proprietary NIS-Elements Advanced Research software (Nikon). Multiple *z*-stacks (~1.07 μm depth) per well from triplicate wells per each bacterial biofilm was collected and three-dimensional images (*xy*, *xz* and *yz* planes) were generated using Nikon software. Calibration was created for both dyes using the software and sequentially collected frames of individual channels were merged and saved as TIFF files.

## Flow cytometry

To determine percentages of live and dead bacteria in the biofilms, two-color flow cytometry assay was performed using a BD FACSymphony™ A5 flow cytometer (BD Biosciences, San Jose, CA). *M. haemolytica* D153 and *S. aureus* NB305 biofilms produced in Immulon 2HB 96-well plates were stained with Filmtracer™ LIVE/DEAD™ Biofilm Viability Kit (Cat#: L10316; ThermoFisher Scientific) as described previously under CLSM. Bacteria was visualized in forward and side light scatter and electronic gates were set to contain bacteria without clumps. Single (SYTO 9 or PI) and double fluorescent dye labeled live and dead bacteria, were included to optimize acquisition gates and compensation for each fluorescent stain. SYTO 9 was excited with a 488 nm laser beam and PI was excited with a 561 nm laser beam and the

emission signals were detected using 530/30 nm and 610/20 nm bandpass filters, respectively. Approximately 10,000 events were collected. Relative percentages of live/dead bacteria in the biofilms were determined using FlowJo software (FlowJo LLC, Ashland, OR).

## Scanning electron microscopy (SEM)

Falcon® 24-well clear, flat bottom tissue culture-treated polystyrene multiwell cell culture plates (Cat# 353047; Becton and Dickinson Co., Franklin Lake, NJ) were used for biofilm study by SEM. Twelve millimeter diameter (#1 thickness, 0.13–0.16 mm) German glass round coverslips (Cat#: 72196–12; Electron Microscopy Sciences (EMS), Hatfield, PA) were carefully placed into each well in the 24-well cell culture plates and 800 μl of the diluted *M. haemolytica* D153 (~1 × 10⁶ cfu/ml in RPMI 1640 medium) and *S. aureus* NB305 (~1 × 10⁶ cfu/ml in BHI broth) cultures were separately added into individual wells and plates were covered with lids. *M. haemolytica* was incubated at 37°C in a humidified atmosphere of 7.5% $CO_2$ for 72 h without shaking, while *S. aureus* was incubated at 37°C for 48 h without shaking, for biofilms to develop on the glass coverslips. Culture media were aspirated and 1 ml of fixative solutions, 10% formalin (NBF; Cat#: 130; Anatech Ltd., Battle Creek, MI), 2.5% glutaraldehyde (Cat#: 16210; EMS) in 0.1M sodium cacodylate buffer, or Methacarn solution (60% methanol (Cat#: 5370–05; Macron Fine Chemicals, Center Valley, PA), 30% chloroform (Cat#: JT9180-3; Mallinckrodt Baker Inc., Phillipsburg, NJ), and 10% glacial acetic acid (Fisher Scientific)) were added into each well. Biofilms were fixed with the individual fixative solutions at room temperature for at least 48 h.

Following fixation, biofilms on the glass coverslips were rinsed with 0.1 M cacodylate buffer (pH 7.2), post-fixed in 1% osmium tetroxide (Cat#: 19170; EMS) in 0.1 M cacodylate buffer for 30 min, rinsed in water (four times, 5 min each), 1% thiocarbohydrazide (once for 20 min), in water (four times, 5 min each), 1% osmium tetroxide (once for 20 min), in water (four times, 5 min each), and dehydrated in a graded series of ethanol (Cat#: 111000190; Pharmco, Richardson, TX; 30% (5 min), 50% (10 min), 70% (two times, 15 min each), 95% (two times, 20 min each), and 100% (two times, 20 min each)). After dehydration in alcohol, samples were chemically dried twice with hexamethyldisilazane (HMDS: Cat#16710; EMS) in ethanol/HMDS (1:1, 20 min), ethanol/HMDS (1:3, 20 min), and HMDS (20 min).

Twelve millimeter conductive carbon adhesive tabs (Cat#: 77825–12; EMS) were attached to Hitachi M4 aluminum specimen mount stubs (Cat#: 16324; Ø15 × 6 mm; TED Pella Inc., Redding, CA) and glass round coverslips with biofilms were carefully attached onto the top. To reduce the arc and further adhere the sample to the stub, colloidal silver liquid (Cat#: 12630; EMS) was applied to the edges between glass coverslips and mount stubs. Biofilms were sputter coated with gold palladium (Cat#: 16771563; Leica Microsystems Inc., Buffalo Grove, IL) for 40 sec using a low vacuum sputter coater (Leica EM ACE200; Leica Microsystems Inc.). *M. haemolytica* and *S. aureus* biofilms were viewed and imaged using a tabletop SEM (Hitachi TM3030 Plus; Hitachi High-Technologies Corporation, Tokyo, Japan). SEM for *M. haemolytica* and *S. aureus* biofilms were repeated two times with triplicate wells (glass coverslips) per each fixative solution.

## Statistical analysis

Biofilm biomass of *M. haemolytica* and *S. aureus* grown in BHI broth and RPMI 1640 medium were determined spectrophotometrically ($OD_{630}$) and presented as mean absorbance with corresponding standard deviation. Live and dead bacteria in the biofilms were determined by flow cytometry after staining with SYTO 9 and propidium iodide and presented as mean percentages with corresponding standard deviation, respectively. Student's *t*-test was used to

compare biomass of biofilms between the two bacteria and between the two culture media as well as to compare percentages of live/dead bacteria in the biofilms. The term significant indicates *P* value of less than 0.05.

## Results and discussion

Several *M. haemolytica* strains including D153 are known to form biofilms *in vitro* on abiotic polystyrene plastic surfaces [24]. Authors reported that *M. haemolytica* forms more biofilm at 37˚C as compared to cattle core body temperature (39˚C), and RPMI 1640 medium was favored over tryptic soy broth (TSB) or RPMI 1640 medium supplemented with 10% fetal bovine serum [24]. It has also been described that *M. haemolytica* forms biofilms on respiratory epithelial cells under *in vitro* culture assay conditions [25]. In healthy animals, palatine tonsilar crypts of the upper respiratory tract serve as a preferred colonization site of *M. haemolytica* [6, 40]. One case study reported the presence of biofilm-like *M. haemolytica* microcolonies in bacterial glycocalyx from a BRDC affected bovine lung sample [26]. However, it is unclear whether long-term colonization of *M. haemolytica* in tonsillar crypts is due to the formation of biofilms or some other unknown mechanism. Various *S. aureus* bovine mastitis isolates including Newbould 305 also form biofilms *in vitro* in TSB and milk serum media and also on epithelial cell cultures [9, 27, 28]. Although it has not been shown *in vivo*, recurrent and chronic mastitis caused by *S. aureus* in cows have been predicted to form biofilms within the mammary glands [15, 29].

In order to confirm that bacteria can produce biofilms on abiotic polystyrene plastic surfaces under our experimental assay conditions (before identifying the best fixative to preserve the biofilms for SEM assessment), we elected to perform *in vitro* quantification of biofilm biomass. Crystal violet stain has been previously used to measure the adherence of coagulase-negative *Staphylococcus* to plastic tissue culture plates [41]. Crystal violet stains EPS, viable, and dead bacteria and therefore this dye has been extensively used to determine the total attached biomass of biofilms. *S. aureus* Newbould 305 (NB305) and *M. haemolytica* (D153) were grown statically in 24-well polystyrene plastic plates for 48 and 72 h, respectively, and biofilms produced were stained with crystal violet which was then dissolved in glacial acetic acid. As expected, robust biofilm formation by both *M. haemolytica* and *S. aureus* were clearly visible with both culture media (Fig 1). Compared to RPMI 1640 medium, both bacteria produced significantly higher amounts of biofilms in BHI broth as indicated by higher absorbance readings at $OD_{630}$ (Fig 1A, *P* <0.0001). Regardless of media, *S. aureus* produced higher amounts of biofilms as compared to *M. haemolytica* (Fig 1B). It is known that various strains of *S. aureus* of human and animal origin, including Newbould 305, can produce biofilms *in vitro* [9, 34, 42]. The Newbould 305 strain has been frequently used by our lab and others to induce chronic mastitis in cattle [11, 43, 44]. However, whether this strain can produce biofilms *in vivo* in the mammary glands of cows is yet to be determined. It has been previously reported that *M. haemolytica* produced more biofilms in RPMI 1640 medium as compared to TSB broth [24]. Although TSB broth was not used in this study, *M. haemolytica* produced significantly higher amounts of biofilms with BHI broth as compared to RPMI 1640 medium (Fig 1B; *P* <0.0001). We routinely use BHI broth as a culture medium to propagate *S. aureus* and *M. haemolytica*. Therefore, production of higher biofilm biomass by these two bacterial species in BHI broth might be attributed to their adaptation to this culture medium and/or availability of higher amount of nutrients in BHI broth as compared to RPMI 1640 medium. Nonetheless, these findings confirmed that *S. aureus* and *M. haemolytica* were able to produce robust biofilms on abiotic polystyrene surfaces under the *in vitro* assay conditions used in the current study.

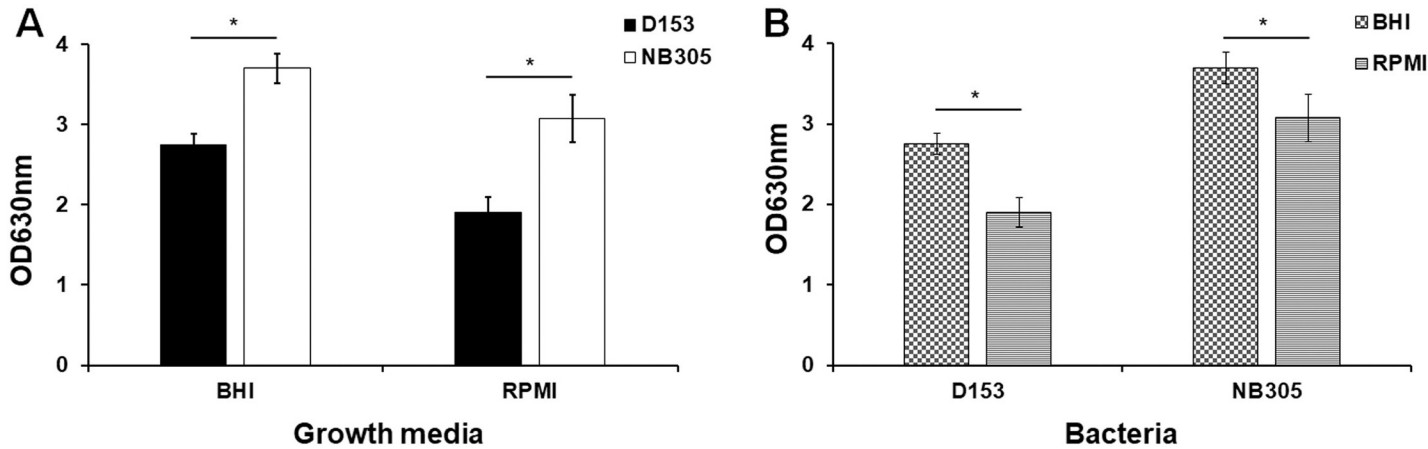

**Fig 1. Quantification of total biomass of biofilms.** *M. haemolytica* (D153) and *S. aureus* Newbould 305 (NB305) were grown in BHI broth and colorless RPMI 1640 in 24-well plates at 37˚C for 48 h (NB305) to 72 h (D153). The effects of growth media (A) and bacteria (B) on the amount of biofilm productions were compared with each other. Biofilms were stained with crystal violet and the absorbance of released crystal violet in glacial acetic acid was measured at 630 nm ($OD_{630}$). Bars indicate mean and standard deviation of total biomass of biofilms from three independent experiments performed in six replicate wells are shown. Differences between biomass of biofilms were determined by Student's *t*-test. * $P < 0.0001$.

Confocal laser scanning microscopy (CLSM) has been successfully used to study the three-dimensional structure of biofilms as well as EPS or matrix composition [45, 46]. CLSM can also be used to quantitatively assess the biovolume, biomass, and thickness of biofilms [47]. However, no attempts were made to quantify biofilms by CLSM in the present work since it is beyond the scope of this study. SYTO 9 and PI, which are two major components in the Film-Tracer LIVE/DEAD biofilm viability kit, can separately detect live and dead [48]. Forty-eight-hours and 72-hours old biofilms of *S. aureus* and *M. haemolytica* grown in Immulon 2HB 96-well clear flat-bottom polystyrene plastic plates and Ibidi 96-well optical bottom (polymer coverslip) μ-plates under static growth conditions were stained with SYTO 9 and PI, and imaged using CLSM. Three-dimensional structure of biofilms assembled with collected Z-stack images of *M. haemolytica* and *S. aureus* in each medium is shown in Fig 2. Both bacterial species grown in BHI medium appeared to form dense and uniform biofilms (Fig 2A, 2C, 2E, and 2G). In contrast, less dense biofilms were formed as evidenced by the presence of bare spots lacking biofilm, when the bacteria were grown in RPMI 1640 medium (Fig 2B, 2D, 2F, and 2H). Nonetheless, CLSM images clearly suggested that both bacterial species produced multilayer biofilms. Although dead bacterial clusters were found in the interior and exterior of the biofilms, as indicated by PI staining, most of the bacteria in the biofilms appeared to be viable as shown by strong SYTO 9 staining (Fig 2). Taken together, findings from the CLSM along with those of quantification of biofilm biomass clearly demonstrate that both *M. haemolytica* and *S. aureus* produce complex multilayer, three-dimensional structure of biofilms under the assay conditions employed in the current study.

Since it was difficult to quantify the percentages of live and dead bacteria in the biofilms by CLSM, a two-color flow cytometry assay was performed. Forty-eight-hours and 72-hours old biofilms of *S. aureus* (NB305) and *M. haemolytica* (D153) grown in BHI broth and RPMI 1640 media in Immulon 2HB 96-well plates were stained with SYTO 9 and PI. Percentages of live and dead bacteria in the biofilms under each growth medium, are shown in Table 1. A significant percentage of live *M. haemolytica* in the biofilms was found in RPMI 1640 medium (80.7%) as compared to BHI broth (71.9%; $P = 0.00007$). Similarly, a significant percentage of live *S. aureus* in the biofilms was found in RPMI 1640 medium (96.7%) as compared to BHI

20×        60×

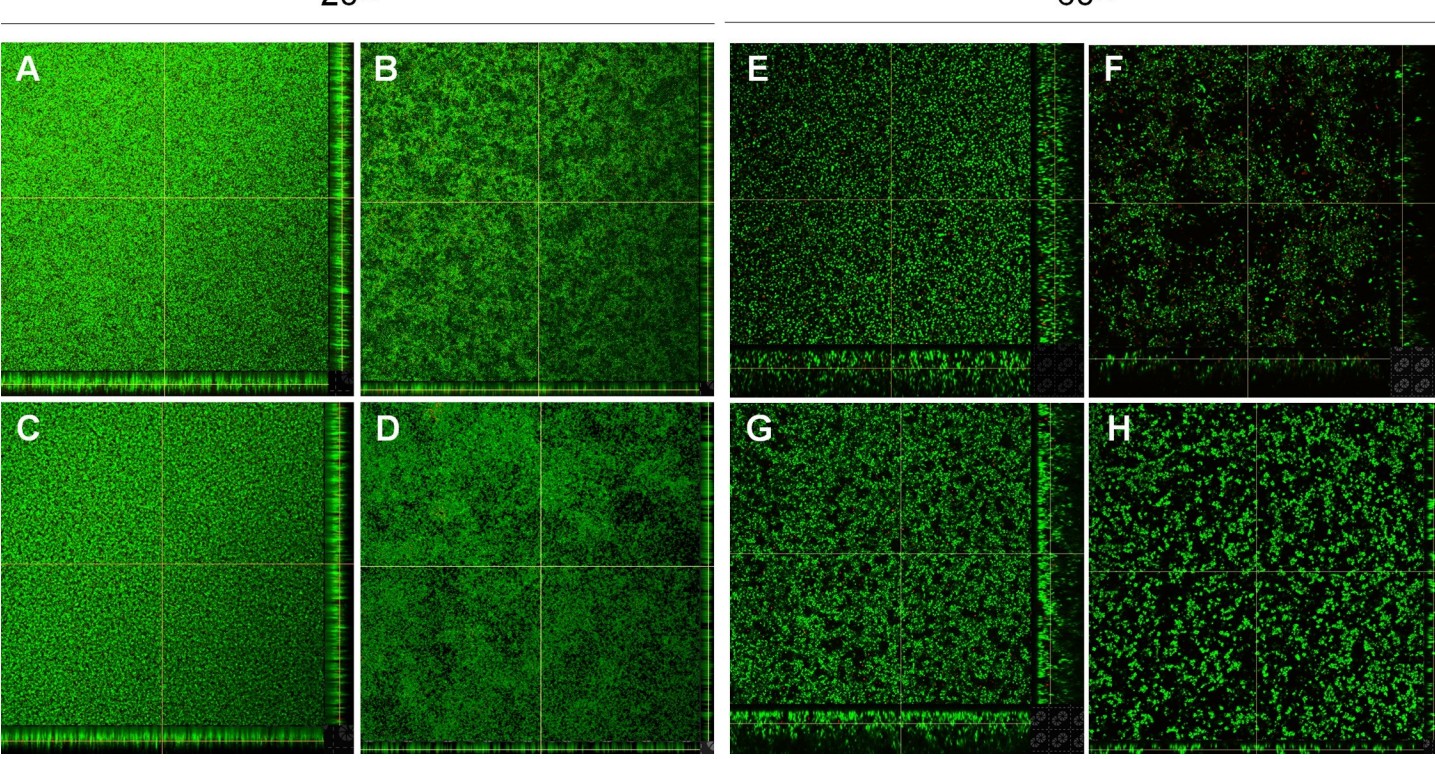

**Fig 2. Confocal laser scanning microscopy of biofilms.** *M. haemolytica* (D153; 20× objective lens: A and B; 60× objective lens: E and F) and *S. aureus* Newbould 305 (NB305; 20× objective lens: C and D; 60× objective lens: G and H) were grown in BHI broth (A, C, E, and G) and colorless RPMI 1640 (B, D, F, and H) in 96-well plates at 37°C for 48 h (NB305) to 72 h (D153). Biofilms grown on polystyrene plastic surfaces (A-D) and polymer coverslips (E-H) were stained with Filmtracer™ LIVE/DEAD™ Biofilm Viability stains (SYTO 9 and propidium iodide) and multiple *z*-stacks were collected per each sample and three-dimensional images were produced using Nikon software. One representative biofilm image out of two independent experiments performed in triplicate wells are shown.

broth (65.9%; *P* < 0.00001). Taken together, findings from the flow cytometry analyses suggest that significant numbers of *M. haemolytica* and *S. aureus* in the biofilms are alive whether bacteria was grown in BHI broth or RPMI 1640.

The major advantage of CLSM over SEM is that biofilm samples do not need to be fixed before or after staining and therefore, fully hydrated and undisturbed biofilms can be visualized and imaged in real-time. If needed, biofilms in the same well can be repeatedly examined for several days and, therefore, kinetics of biofilm development can also be studied. Although the complex three-dimensional structure of biofilms can be produced with Z-stacks, the major disadvantage of CLSM when compared to SEM is that CLSM does not have the high resolution

**Table 1. Flow cytometry results of live and dead bacteria in the biofilms.**

|  | BHI | | RPMI 1640 | |
|---|---|---|---|---|
|  | Live | Dead | Live | Dead |
| *M. haemolytica* (D153) | 71.9 ± 3.3 | 27.2 ± 3.3 | 80.7 ± 1.5 | 19.0 ± 1.5 |
| *S. aureus* (NB305) | 65.9 ± 1.8 | 34.1 ± 1.8 | 96.7 ± 1.2 | 3.3 ± 1.2 |

*M. haemolytica* (D153) and *S. aureus* Newbould 305 (NB305) were grown in BHI broth and RPMI 1640 in 96-well plates at 37°C for 48 h (NB305) to 72 h (D153). Bacteria in the biofilms were stained with Filmtracer™ LIVE/DEAD™ Biofilm Viability stains and the viability was determined by flow cytometry analysis. Mean percentages of live and dead bacteria with corresponding SD are shown.

and magnification power to capture detailed cellular morphology of bacteria and EPS structures in the biofilms. However conventional SEM requires specimen treatment with an appropriate fixative solution. If a suitable fixative has not been selected to fix the sample, biofilms can collapse and/or be completely destroyed beyond recognition.

Although complex and multilayer biofilm structure formation has been confirmed by crystal violet staining and CLSM for various bacterial species, SEM images often show poor quality EPS structures [33–35, 49]. Most of these studies used formalin (NBF) or glutaraldehyde fixative to fix bacterial biofilms [25, 33–35]. It has been previously reported that mucus and mucosal flora in the adenoid tissues and gut tissues can be preserved using Carnoy's fixative [36, 37]. Furthermore, authors suggested that use of nonaqueous Carnoy's fixative (60% ethanol, 30% chloroform, and 10% glacial acetic acid) [50] was crucial to preserve the bacteria attached to mucus layer and biofilms which were not detected in formalin-fixed gut biopsies [36]. Methacarn (Methanol-Carnoy) fixative is very similar to Carnoy's fixative with the exception of methanol instead of ethanol (60% methanol, 30% chloroform, and 10% glacial acetic acid) [51]. It is our understanding that neither Carnoy's nor Methacarn fixative solution has been widely tested with bacterial biofilms produced under *in vitro* assay conditions for SEM analysis. Therefore, the objective of this study was to identify a suitable fixative solution that can preserve the complex structure of biofilms in a life-like state. Forty-eight-hours and 72-hours old biofilms of *S. aureus* NB305 and *M. haemolytica* D153 grown on round glass coverslips under static growth conditions were fixed with individual fixative for at least 48 h at 4˚C and then all the samples were similarly processed for SEM. As expected, all three fixative solutions were effective in preserving the bacterial cell morphology such as coccobacilli (*M. haemolytica*; Fig 3A, 3B and 3C) and cocci (*S. aureus*; Fig 3D, 3E and 3F). However, in representative fields of view, only remnants of EPS were found with NBF (Fig 3A–3D) and glutaraldehyde fixatives (Fig 3B–3E). Total loss of bacteria and biofilms have also been reported for NBF-fixed gut biopsies [36]. Although, we also observed a partial loss of bacteria and biofilms on glass coverslips with NBF, minimal losses were observed with glutaraldehyde. However, collapsed EPS along with densely packed bacteria were clearly visible with glutaraldehyde (Fig 3B–3E). Similar to our findings, careful analysis of previously published work on *S. aureus* biofilms, which were fixed in glutaraldehyde, also showed collapsed EPS [52]. Unlike both NBF- and glutaraldehyde-fixed biofilms, well preserved EPS in both *M. haemolytica* (Fig 3C and S1A Fig) and *S. aureus* (Fig 3F and S1B Fig) biofilms were readily observed with Methacarn fixative. EPS layers were clearly visible on the top, in the middle and in the bottom of both biofilms (Fig 3C–3F, indicated by arrows). As expected, single bacterium and bacterial clusters (microcolonies) attached to EPS layers were also observed. Compared to *M. haemolytica* biofilm (Fig 3C and S1A Fig), EPS layers in *S. aureus* biofilm appeared to be in a uniform film-like format (Fig 3F and S1B Fig). Although uniform film-like format EPS layers were also visible with *M. haemolytica* biofilm, string-like EPS structures were also observed (Fig 3C). Relatively thicker levels of EPS layers were observed for *S. aureus* biofilm as compared to *M. haemolytica* biofilm. This difference might be attributed to the use of nutrient-rich BHI broth for *S. aureus* as compared to RPMI 1640 medium for *M. haemolytica* to produce biofilms, or *S. aureus* is capable of producing more complex biofilms than *M. haemolytica* as it has been observed with crystal violet staining (Fig 1).

It is known that fixatives can induce shrinkage or swelling of tissues and therefore, a universal fixative which could fit for all occasions has not been identified yet. Therefore, the goal of this study was to identify a fixative solution that can preserve both bacterial cell morphology as well as EPS to visualize complex architecture of bacterial biofilms under SEM. Cross-linking fixatives induce formation of inter- and intra-molecular cross-links between proteins and nucleic acids [30]. Cross-linking fixatives such as formalin do not alter the secondary structure

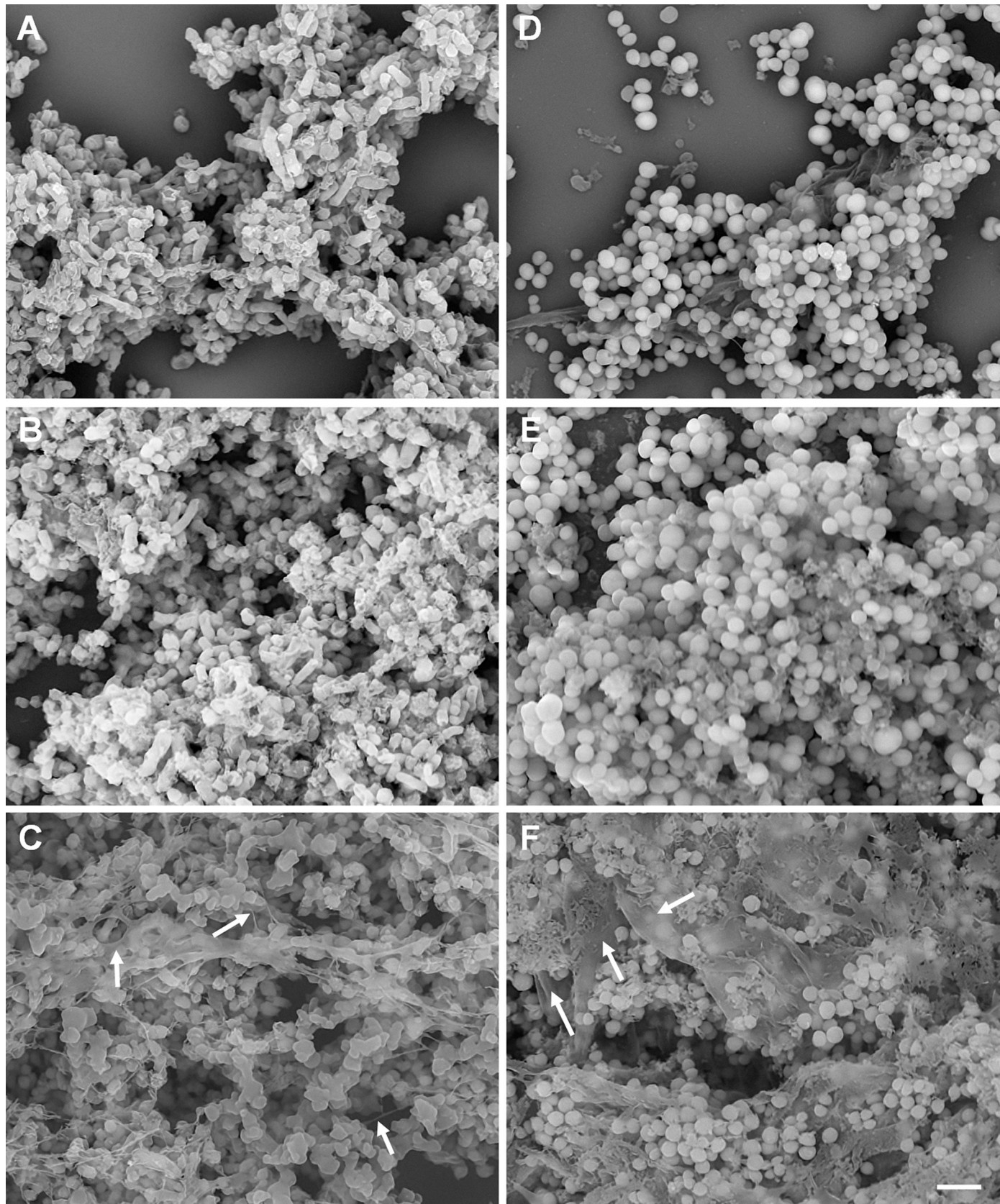

**Fig 3. Scanning electron microscopy of biofilms.** *M. haemolytica* (D153), grown in colorless RPMI 1640 (A, B, and C) and *S. aureus* Newbould 305 (NB305), grown in BHI broth (D, E, and F) on round glass coverslips in 24-well plates at 37°C for 48 h (NB305) to 72 h (D153). Biofilms grown on glass coverslips were fixed with 10% formalin (A and D), 2.5% glutaraldehyde (B and E), or Methacarn (C and F) fixative solutions for 48 h and samples were further processed for SEM examination. EPS layers on the top, in the middle, and in the bottom of biofilms (C and F) are shown by white arrows. One representative SEM biofilm image out of two independents experiments performed in triplicate wells are shown. (Bar = 4 μm; 5,000 magnification).

of proteins when in liquid [53]. In contrast, alcohol fixatives, such as ethanol and methanol are known to remove free and bound water molecules causing changes in the tertiary structure of proteins and promoting protein precipitation and tissue shrinkage. However, alcohol fixation does not cause detectable changes to the nucleic acids. It has been reported that the addition of the mixture of acetic acid and chloroform will prevent alcohol-induced shrinkage effects on the tissue [30]. Since methanol is structurally closer to water than ethanol, methanol competes less strongly than ethanol for hydrophobic areas. Therefore, we selected Methacarn fixative over Carnoy's fixative to fix biofilms for SEM evaluation. Since Carnoy's solution has been successfully used to preserve bacteria and biofilms on mucus membranes [36, 37], we were confident that Carnoy's fixative should work very similar to Methacarn fixative when preserving bacterial biofilms formed at the solid-liquid interface on abiotic surfaces. Based on the findings in this study, we suggest that Methacarn (and Carnoy's) fixative solution is the best fixative solution to preserve the complex architecture of biofilms formed at the solid-liquid interface and to visualize detailed structures of EPS layers for SEM evaluation.

## Supporting information

**S1 Fig. Scanning electron microscopy of biofilms.** *M. haemolytica* (D153), grown in colorless RPMI 1640 (A) and *S. aureus* Newbould 305 (NB305), grown in BHI broth (B) on round glass coverslips in 24-well plates at 37°C for 48 h (NB305) to 72 h (D153). Biofilms grown on glass coverslips were fixed with Methacarn fixative solutions for 48 h and samples were further processed for SEM examination. One representative SEM biofilm image out of two independents experiments performed in triplicate wells are shown. (Bar = 4 μm; 12,000 magnification). (TIF)

## Acknowledgments

The authors would like to thank Tracy Porter, Duane Zimmerman, Sam Humphrey, and Brad Chriswell at the NADC for their excellent technical support. We would also like to thank Drs. Marcus E. Kehrli Jr and Vijay K. Sharma at the NADC for helpful discussions and critical review of the manuscript.

**Disclaimer:** Mention of trade names or commercial products in this article is solely for the purpose of providing specific information and does not imply recommendation or endorsement by the U.S. Department of Agriculture. USDA is an equal opportunity provider and employer.

## Author Contributions

**Conceptualization:** Rohana P. Dassanayake.

**Data curation:** Rohana P. Dassanayake.

**Formal analysis:** Rohana P. Dassanayake, Shollie M. Falkenberg, Judith A. Stasko, Adrienne L. Shircliff.

**Funding acquisition:** Rohana P. Dassanayake, John D. Lippolis, Robert E. Briggs.

**Investigation:** Rohana P. Dassanayake, Shollie M. Falkenberg, Judith A. Stasko, Adrienne L. Shircliff, John D. Lippolis, Robert E. Briggs.

**Methodology:** Rohana P. Dassanayake, Judith A. Stasko.

**Project administration:** Rohana P. Dassanayake.

**Resources:** Rohana P. Dassanayake, Shollie M. Falkenberg, Judith A. Stasko, Adrienne L. Shircliff, John D. Lippolis, Robert E. Briggs.

**Supervision:** Rohana P. Dassanayake.

**Validation:** Rohana P. Dassanayake, Judith A. Stasko, Adrienne L. Shircliff.

**Visualization:** Rohana P. Dassanayake, Judith A. Stasko, Adrienne L. Shircliff.

**Writing – original draft:** Rohana P. Dassanayake.

**Writing – review & editing:** Rohana P. Dassanayake, Shollie M. Falkenberg, Judith A. Stasko, Adrienne L. Shircliff, John D. Lippolis, Robert E. Briggs.

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
