## [Decision Letter · Decision Letter 0]

8 Apr 2020

PONE-D-20-07233

Identification of a reliable fixative solution to preserve the complex architecture of bacterial biofilms for scanning electron microscopy evaluation

PLOS ONE

Dear Dr. Dassanayake,

Thank you for submitting your manuscript to PLOS ONE. After careful consideration, we feel that it has merit but does not fully meet PLOS ONE’s publication criteria as it currently stands. Therefore, we invite you to submit a revised version of the manuscript that addresses the points raised during the review process.

We would appreciate receiving your revised manuscript by May 23 2020 11:59PM. To enhance the reproducibility of your results, we recommend that if applicable you deposit your laboratory protocols in protocols.io, where a protocol can be assigned its own identifier (DOI) such that it can be cited independently in the future. For instructions see: http://journals.plos.org/plosone/s/submission-guidelines#loc-laboratory-protocols

We look forward to receiving your revised manuscript.

Kind regards,

Amitava Mukherjee, ME, Ph.D.

Academic Editor

PLOS ONE

Reviewers' comments:

Reviewer's Responses to Questions

**Comments to the Author**

1. Is the manuscript technically sound, and do the data support the conclusions?

Reviewer #1: Partly

2. Has the statistical analysis been performed appropriately and rigorously? 

Reviewer #1: Yes

3. Have the authors made all data underlying the findings in their manuscript fully available?

Reviewer #1: Yes

4. Is the manuscript presented in an intelligible fashion and written in standard English?

Reviewer #1: Yes

5. Review Comments to the Author

Reviewer #1: 1. Is the manuscript technically sound, and do the data support the conclusions?

The manuscript attempts to find the ability of Mannheimia haemolytica (D153) and Staphylococcus aureus (NB305) to form 'better biofilms and EPS' in Methacarn fixative solution. The observations came from crystal violet staining method, confocal laser scanning microscopy and scanning electron microscopy.

# Figure 1:

a. What are the 'specific' reasons to see increased absorbance of NB305 and D153 when grown in BHI broth (brain heart infusion) over RPMI medium?

b. Crystal violet stains dead bacteria as well, and so quantification of biofilm biomass might include the dead bacteria along with live ones and EPS. Is there any alternate methods available that can better-quantify the biofilm?

c. How about the influence of media alone (without bacteria) for absorbance?

d. In Fig.1 'legend', please provide description for A) and B). At present it is a full paragraph and the figure shows A and B.

# Figure 2:

a. NB305 and D153 grown in BHI medium appeared to form dense and uniform biofilms. Please include a subset to show a close-up view of a stack for clarity. At present, it looks over-crowded and not able to discern any feature. These images are 3D right.

b. Same goes here. Though the colonies are crowded in pockets, still discerning the details is daunting.

c. As authors indicated using CLSM to quantify the biovolume, biomass, and thickness of biofilms is beyond the reach of the study. In the absence of much data, these quantifications might strengthen the finding.

d. At least, showing the quantitative count on dead and viable colonies to form biofilm using Live tracker of CLSM might be of help.

# Figure 3:

SEM images to show the biofilm in various fixatives. Better visibility of EPS in methacarn fixative. Please choose a part of an image and zoom to show the feature of EPS in the methacarn treated sample.

Limitations of the study:

a. Whether these strains can produce biofilms in vivo was not tested?

b. Limited data for a full-length research article.

2. Has the statistical analysis been performed appropriately and rigorously?

This study makes use of Student's t-test and not beyond. This is sufficient.

3. Have the authors made all data underlying the findings in their manuscript fully available?

Figures are available.

4. Is the manuscript presented in an intelligible fashion and written in standard English?

Yes, the manuscript has been written in standard English.

6. PLOS authors have the option to publish the peer review history of their article (what does this mean?). If published, this will include your full peer review and any attached files.

Reviewer #1: No

---

## [Author Response · Author response to Decision Letter 0]

27 Apr 2020

Comment:

# Figure 1:

a. What are the 'specific' reasons to see increased absorbance of NB305 and D153 when grown in BHI broth (brain heart infusion) over RPMI medium?

Response:

As we described in the manuscript at line# 276-279, we suggest that adaptation of both bacteria to BHI broth (since we routinely grow both bacteria in BHI broth) and higher nutrients in BHI broth (as compared to RPMI medium) might have contributed to higher bacterial growth and thus higher biofilm productions. 

Comment:

b. Crystal violet stains dead bacteria as well, and so quantification of biofilm biomass might include the dead bacteria along with live ones and EPS. Is there any alternate methods available that can better-quantify the biofilm?

Response:

Yes, in addition to crystal violet stain, other stains such as SYTO 9, fluorescein diacetate, resazurin, XTT, and dimethyl methylene blue assays can also be used to quantify biofilms. However, standard method to quantify biofilms is still the staining with the crystal violet. The major advantage of crystal violet is that it not only stains bacteria (both live and dead) but also EPS. Therefore, we used crystal violet staining method to quantify both bacterial biofilms grown in both types of growth media. 

However, if the goal of the experiment is to determine the killing of bacteria in biofilms after certain treatments, then the crystal violet is not a good choice since it stains both live and dead bacteria.

Comment:

c. How about the influence of media alone (without bacteria) for absorbance?

d. In Fig.1 'legend', please provide description for A) and B). At present it is a full paragraph and the figure shows A and B.

Response:

(c). We used media (without bacteria) as a negative controls (line# 136-137). These wells treated similar to biofilms containing wells during fixing, staining with crystal violet, washing and then incubation with 33% acetic acid. The absorbance of media control wells are very similar to empty wells and therefore media alone wells did not influence absorbance, but the absorbance was influenced only by the amount of biofilms. 

(d). We apologize for not including the description of A and B. The description of A and B [“The effects of growth media (A) and bacteria (B) on the amount of biofilm productions were compared with each other.”] is now added to the Fig.1 legend (line# 594-595). 

Comment:

# Figure 2:

a. NB305 and D153 grown in BHI medium appeared to form dense and uniform biofilms. Please include a subset to show a close-up view of a stack for clarity. At present, it looks over-crowded and not able to discern any feature. These images are 3D right.

Response:

(a). Since we used Immulon 2HB plates, we could only collect images at 20× objective lens. Taking reviewer’s suggestion into consideration, we conducted new experiments to produce biofilms for both bacteria using Ibidi 96 well µ-plates (line# 156-157). Since these plates have “optical bottoms/polymer coverslips”, we were able to collected images using 60× objective lens (line# 171-172). The 60× objective lens images are now added to the revised Fig. 2 E-H. 

Yes, these are 3D images of the biofilms. 

Comment:

b. Same goes here. Though the colonies are crowded in pockets, still discerning the details is daunting.

Response:

(b). Please see the previous response. 60× objective lens are included in Fig. 2 E-H.

Comment:

c. As authors indicated using CLSM to quantify the biovolume, biomass, and thickness of biofilms is beyond the reach of the study. In the absence of much data, these quantifications might strengthen the finding.

Response:

(c). We agree with reviewers comment. The goal of current study was to identify a reliable fixative for biofilms for SEM evaluation. However, we will take reviewer’s suggestion into consideration in our future studies. 

Comment:

d. At least, showing the quantitative count on dead and viable colonies to form biofilm using Live tracker of CLSM might be of help.

Response:

(d) Our CLSM is not equipped with an appropriate module to quantify live and dead bacteria in the biofilms after SYTO 9/PI staining. Therefore, we have conducted new experiments to determine the percent live/dead bacteria using flow cytometry technique after staining biofilms with SYTO 9 and PI. This new method (line# 179-191) and results/discussion (line# 303-313) are now added to the revised manuscript. The percentages of live and dead bacteria with corresponding standard deviations are shown in Table 1 (line# 621). 

Comment:

# Figure 3:

SEM images to show the biofilm in various fixatives. Better visibility of EPS in methacarn fixative. Please choose a part of an image and zoom to show the feature of EPS in the methacarn treated sample.

Response:

Since we have used Hitachi Table Top SEM, it does not allow us to take higher magnification images. The maximum magnification without distorting image we can collect is at 12,000 ×. Although we try to insert a selected area of EPS, it did not provided additional information. Therefore taken reviewer’s recommendation into consideration, we prepared a supplementary figure (Fig. S1) and figure legends (line# 629-635). 

Comment:

Limitations of the study:

a. Whether these strains can produce biofilms in vivo was not tested?

b. Limited data for a full-length research article.

Response:

(a) We agree with reviewer that it is important to determine whether M. haemolytica and S. aureus are able to produce biofilms in vivo. However, before such studies it is important to identify a superior fixative solution to fix both bacteria and EPS in the biofilms. Since now we know that methacarn fixative solution is superior to preserve EPS (and bacteria) in the biofilms in vitro as compared to NBF and glutaraldehyde, we will use this fixative in future for in vivo biofilm studies. 

(b) We respectfully disagree with this assessment. We have conducted several techniques such as crystal violet staining, CLSM, and flow cytometry in addition to SEM to characterize both M. haemolytica and S. aureus grown in BHI and RPMI media. Therefore, we strongly believe that multiple techniques along with findings of methacarn as a superior fixative to preserve EPS and bacteria in the biofilms warrant this study for a full-length research article. 

Comment:

2. Has the statistical analysis been performed appropriately and rigorously?

This study makes use of Student's t-test and not beyond. This is sufficient.

Response:

We highly appreciate reviewer’s assessment. 

Comment:

3. Have the authors made all data underlying the findings in their manuscript fully available?

Figures are available.

Response:

We highly appreciate reviewer’s assessment. 

Comment:

4. Is the manuscript presented in an intelligible fashion and written in standard English?

Yes, the manuscript has been written in standard English.

Response:

We highly appreciate reviewer’s assessment.

---

## [Decision Letter · Decision Letter 1]

18 May 2020

Identification of a reliable fixative solution to preserve the complex architecture of bacterial biofilms for scanning electron microscopy evaluation

PONE-D-20-07233R1

Dear Dr. Dassanayake,

We are pleased to inform you that your manuscript has been judged scientifically suitable for publication and will be formally accepted for publication once it complies with all outstanding technical requirements.

With kind regards,

Amitava Mukherjee, ME, Ph.D.

Academic Editor

PLOS ONE

Additional Editor Comments (optional):

Reviewers' comments:

Reviewer's Responses to Questions

**Comments to the Author**

1. If the authors have adequately addressed your comments raised in a previous round of review and you feel that this manuscript is now acceptable for publication, you may indicate that here to bypass the “Comments to the Author” section, enter your conflict of interest statement in the “Confidential to Editor” section, and submit your "Accept" recommendation.

Reviewer #1: All comments have been addressed

2. Is the manuscript technically sound, and do the data support the conclusions?

Reviewer #1: Yes

3. Has the statistical analysis been performed appropriately and rigorously? 

Reviewer #1: Yes

4. Have the authors made all data underlying the findings in their manuscript fully available?

Reviewer #1: Yes

5. Is the manuscript presented in an intelligible fashion and written in standard English?

Reviewer #1: Yes

6. Review Comments to the Author

Reviewer #1: Dassanayake et al. addressed all the queries. Authors have performed new experiments. The intriquing observation is presence of more biofilms observed in CLSM is not represented the same way when quantified in flow cytometry. For example, in flow cytometry the percentage of live bacteria more in RPMI medium than seen in BHI broth (Table 1). When you compare this results with CLSM results, less dense biofilms are seen in RPMI medium (Line No: 294 & 295, Fig.2 F). Results from CLSM and flow cytometry count not going hand-in-hand. May be the field of observation and field of count are different. Kindly check this please.

7. PLOS authors have the option to publish the peer review history of their article (what does this mean?). If published, this will include your full peer review and any attached files.

Reviewer #1: No

---

## [Editor Report · Acceptance letter]

21 May 2020

PONE-D-20-07233R1 

Identification of a reliable fixative solution to preserve the complex architecture of bacterial biofilms for scanning electron microscopy evaluation 

Dear Dr. Dassanayake:

I am pleased to inform you that your manuscript has been deemed suitable for publication in PLOS ONE. Congratulations! Your manuscript is now with our production department. 

With kind regards,

on behalf of

Professor Dr. Amitava Mukherjee 

Academic Editor

PLOS ONE